# RNA secondary structure prediction using deep learning with thermodynamic integration

Kengo Sato [1✉], Manato Akiyama[1] & Yasubumi Sakakibara[1]

Accurate predictions of RNA secondary structures can help uncover the roles of functional non-coding RNAs. Although machine learning-based models have achieved high performance in terms of prediction accuracy, overfitting is a common risk for such highly parameterized models. Here we show that overfitting can be minimized when RNA folding scores learnt using a deep neural network are integrated together with Turner's nearest-neighbor free energy parameters. Training the model with thermodynamic regularization ensures that folding scores and the calculated free energy are as close as possible. In computational experiments designed for newly discovered non-coding RNAs, our algorithm (MXfold2) achieves the most robust and accurate predictions of RNA secondary structures without sacrificing computational efficiency compared to several other algorithms. The results suggest that integrating thermodynamic information could help improve the robustness of deep learning-based predictions of RNA secondary structure.

[1] Department of Biosciences and Informatics, Keio University, 3-14-1 Hiyoshi, Kohoku-ku, Yokohama, Japan. ✉email: satoken@bio.keio.ac.jp

Recent studies have revealed that functional non-coding RNAs (ncRNAs) play essential roles, including transcriptional regulation and guiding modification, participating in various biological processes ranging from development to cell differentiation, with defective functionality being involved in various diseases[1]. Because it is well-known that the functions of ncRNAs are deeply related to their structures rather than their primary sequences, discovering the structures of ncRNAs can elucidate the functions of ncRNAs. However, there are major difficulties in determining RNA tertiary structures through experimental assays such as nuclear magnetic resonance and X-ray crystal structure analysis, because of the high experimental costs and resolution limits on measurements of RNA. Although considerable advances in cryo-electron microscopy research on RNA tertiary structure determination have been achieved in recent years[2], these limitations have not yet been completely overcome. Therefore, instead of conducting such experimental assays, we frequently perform computational prediction of RNA secondary structures, defined as sets of base-pairs with hydrogen bonds between the nucleotides.

The most popular approach for predicting RNA secondary structures is based on thermodynamic models, such as Turner's nearest-neighbor model[3,4], in which a secondary structure is decomposed into several characteristic substructures, called nearest-neighbor loops, such as hairpin loops, internal loops, bulge loops, base-pair stackings, multi-branch loops, and external loops, as shown in Fig. 1. The free energy of each nearest-neighbor loop can be calculated by summing the free energy parameters that characterize the loop. The free energy parameters have been determined in advance by experimental methods such as optical melting experiments[3]. The free energy of an entire RNA secondary structure is calculated by summing the free energy of the decomposed nearest-neighbor loops. We can efficiently calculate an optimal secondary structure that has the minimum free energy using a dynamic programming (DP) technique, the well-known Zuker algorithm[5]. A number of tools, including Mfold/UNAfold[6,7], RNAfold[8,9], and RNAstructure[10,11], have adopted this approach.

An alternative approach utilizes machine learning techniques, which train scoring parameters for decomposed substructures from reference structures, rather than experimental techniques. This approach has successfully been adopted by CONTRAfold[12,13],

ContextFold[14], and other methods, and has enabled us to more accurately predict RNA secondary structures. However, rich parameterization can easily cause overfitting to the training data, thus preventing robust predictions for a wide range of RNA sequences[15]. Probabilistic generative models such as stochastic context-free grammars (SCFGs) have also been applied to predicting RNA secondary structures. Recently, TORNADO[15] implemented an application of SCFGs to the nearest-neighbor model, thus achieving performance comparable with that of other machine learning-based methods.

Hybrid methods that combine thermodynamic and machine learning-based approaches to compensate for each other's shortcomings have been developed. SimFold[16,17] more accurately estimates thermodynamic parameters from training data, including RNA sequences and their known secondary structures, as well as the free energy of their known secondary structures. We previously proposed MXfold[18], which combines thermodynamic energy parameters and rich-parameterized weight parameters. The model learns more precise parameters for substructures observed in training data and avoids overfitting rich-parameterized weight parameters to the training data by resorting to the thermodynamic parameters for assessing previously unobserved substructures.

In recent years, deep learning has made remarkable progress in a wide range of fields, including bioinformatics and life sciences. SPOT-RNA[19] and E2Efold[20] have been developed for RNA secondary structure prediction using deep neural networks (DNNs). Both algorithms formulate RNA secondary structure prediction as multiple binary classification problems that predict whether each pair of nucleotides forms a base pair or not, using a deep neural network trained with a large amount of training data. As with the other machine learning-based methods described above, concerns about overfitting caused by rich parameterization remain.

Inspired by MXfold and the DNN-based RNA secondary structure prediction methods, in this paper, we propose an algorithm for predicting RNA secondary structures using deep learning. Similar to MXfold, we integrate folding scores, which are calculated by a deep neural network, with Turner's nearest-neighbor free energy parameters. The deep neural network is trained using the max-margin framework with thermodynamic regularization, which is proposed in this paper to make our model robust by maintaining that the folding score calculated by our method and the free energy calculated by the thermodynamic parameters are as close as possible. To confirm the robustness of our algorithm, we conducted computational experiments with two types of cross-validation: sequence-wise cross-validation, with test datasets structurally similar to training datasets, and family-wise cross-validation, with test datasets structurally dissimilar to training datasets. The resulting method, called MXfold2, achieved the best prediction accuracy based not only on the sequence-wise cross-validation, but also on the family-wise cross-validation, indicating the robustness of our algorithm. In addition, the computational experiment conducted on a dataset with sequence-structure-energy triplets shows a high correlation between folding scores predicted by MXfold2 and the free energy derived from optical melting experiments.

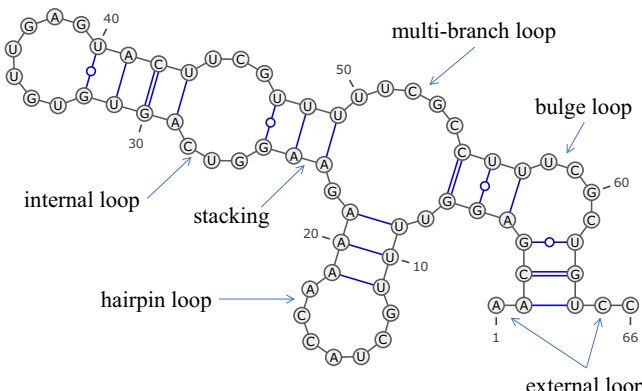

**Fig. 1 Decomposition of an RNA secondary structure into nearest-neighbor loops.** An RNA secondary structure can be decomposed into several types of nearest-neighbor loops, including hairpin loops (e.g., bases 11–19), internal loops (e.g., bases 25–29 and 43–47), bulge loops (e.g., bases 4–5 and 57–62), base-pair stackings (e.g., bases 23–24 and 48–49), multi-branch loops (e.g., bases 7–9, 21–23, and 49–55), and external loops (e.g., bases 1–2 and 64–66). This diagram was drawn using VARNA[45].

## Results

**Overview of our algorithm.** Our algorithm computes four types of folding scores for each pair of nucleotides by using a deep neural network, as shown in Fig. 2. The folding scores are used to calculate scores of the nearest-neighbor loops. Similar to MXfold, we integrate the folding scores, which are calculated using a deep neural network, with Turner's nearest-neighbor free energy

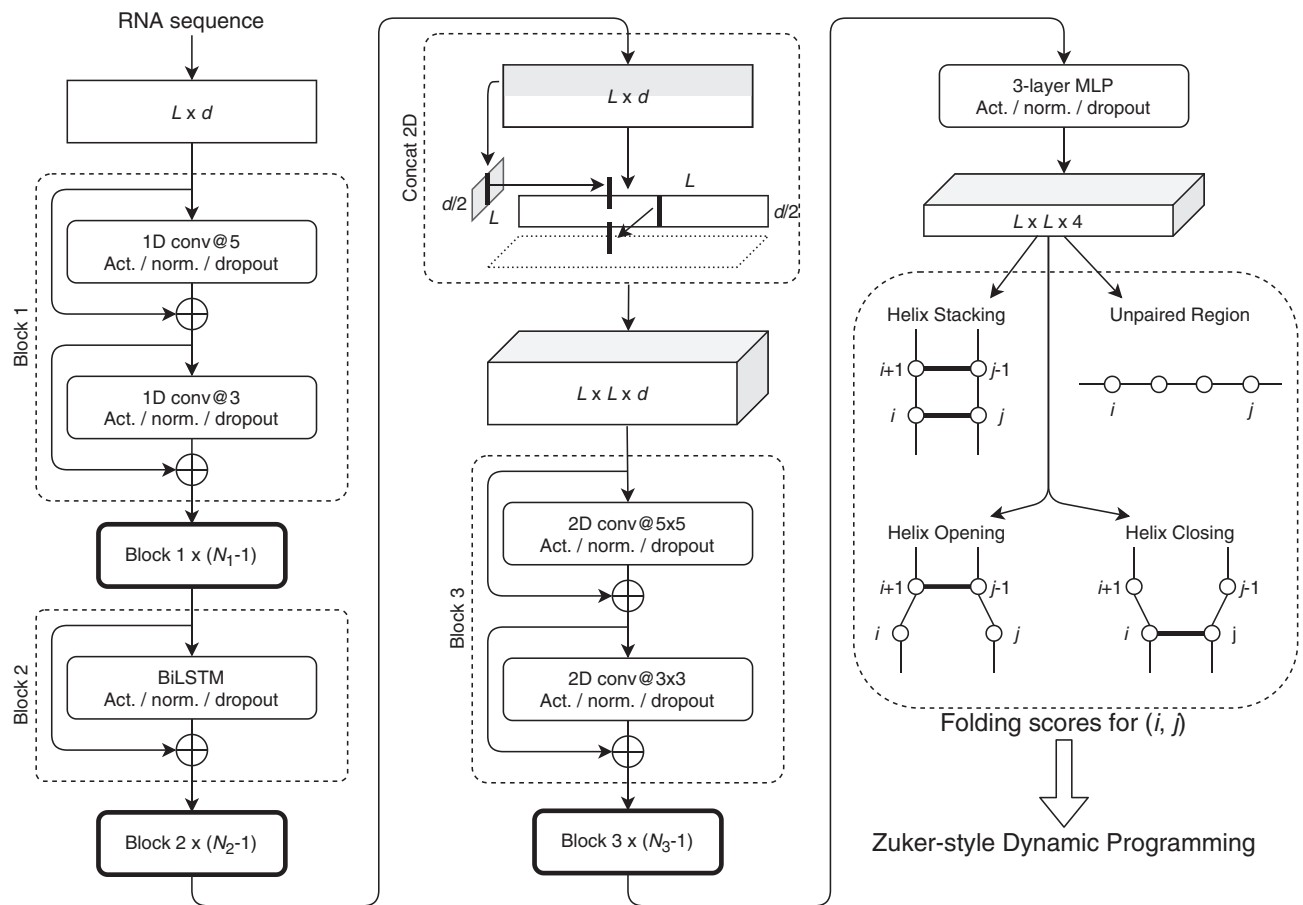

**Fig. 2 The network structure of our algorithm.** It calculates four types of folding scores for each pair of nucleotides $(i, j)$ from a given RNA sequence of length $L$. See the "Methods" section for more details. Act.: the activation function, Norm.: the normalization function, Conv: the convolution layer, $d$: the dimension of trainable embeddings, $N_1$, $N_2$, and $N_3$: the numbers of times Blocks 1, 2, and 3 repeat, respectively, BiLSTM: bidirectional long short-term memory, MLP: multilayer perceptron, and Concat 2D: concatenation of 1D sequences to 2D matrices.

parameters. Then, our algorithm predicts an optimal secondary structure that maximizes the sum of the scores of the nearest-neighbor loops using Zuker-style dynamic programming (DP)[5]. The deep neural network is trained using the max-margin framework, also known as structured support vector machine (SSVM), to minimize the structured hinge loss function with thermodynamic regularization, which prevents the folding score of the secondary structure from differing significantly from the free energy of the thermodynamic parameters.

**Effectiveness of two thermodynamic-related techniques**. To investigate the effectiveness of the thermodynamic-integrated folding scores and thermodynamic regularization, which are the main contribution of this study, we conducted an experiment to compare the cases with and without the thermodynamic-related techniques. We implemented a base model that employs only the deep neural network to compute the folding score; that is, it does not use the thermodynamic parameters for the folding score and the thermodynamic regularization. Then, we compared the base model with the use of the thermodynamic-integrated folding score and/or the thermodynamic regularization. For this experiment, we employed TrainSetA, TestSetA, and TestSetB, which have been established by Rivas et al.[15]. Note that TrainSetA and TestSetB are structurally dissimilar, whereas TrainSetA and TestSetA have some structural similarity. All the models were trained with TrainSetA and were then evaluated using TestSetA and TestSetB. The results in Table 1 and Supplementary Fig. 1

**Table 1 Comparison of the accuracy of the secondary structure prediction between the base model and the use of the thermodynamic-related techniques.**

|  | TestSetA | | | TestSetB | | |
|---|---|---|---|---|---|---|
|  | PPV | SEN | F | PPV | SEN | F |
| Base model | 0.794 | 0.824 | 0.804 | 0.461 | 0.545 | 0.494 |
| +int.[a] | 0.799 | 0.825 | 0.808 | 0.500 | 0.571 | 0.527 |
| +reg.[b] | 0.765 | 0.796 | 0.776 | 0.542 | 0.647 | 0.583 |
| Final model | 0.754 | 0.778 | 0.761 | 0.571 | 0.650 | 0.601 |

[a]The use of the thermodynamic-integrated folding scores.
[b]The use of the thermodynamic regularization.

show that the base model achieved high accuracy for TestSetA, but quite poor accuracy for TestSetB, suggesting that the base model might be categorized as being prone to heavy overfitting. In contrast, the model using the complete thermodynamic-related techniques (the final model) showed much better accuracy for TestSetB compared with the base model, indicating that the two thermodynamic-related techniques enable robust predictions.

**Comparison with the existing methods**. We compared our algorithm with nine available folding algorithms: MXfold version 0.0.2[18], LinearFold-V (committed on Feb 5, 2020)[21], CONTRA-fold version 2.02[12,13], ContextFold version 1.00[14], CentroidFold

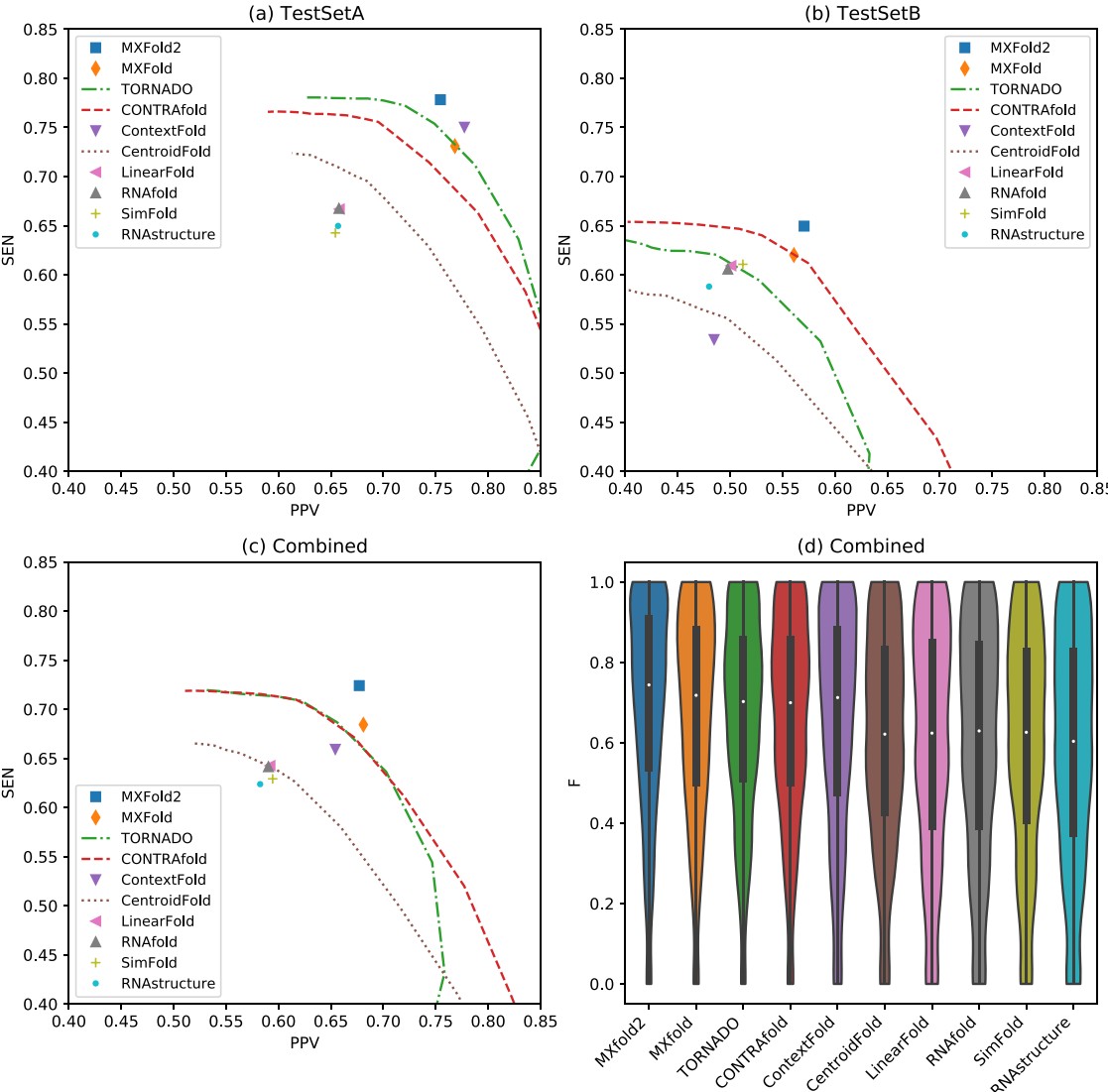

**Fig. 3 Comparison of prediction accuracy with the existing methods.** Positive predictive value (PPV) and sensitivity (SEN) plots for (**a**) TestSetA, (**b**) TestSetB, and (**c**) the combined dataset with TestSetA and TestSetB. (**d**) $F$-values of the combined test data for all the methods. See Supplementary Tables 2–4 and Supplementary Fig. 5 for more details.

version 0.0.16[22,23], TORNADO version 0.3[15] with ViennaRNA grammar, RNAfold in ViennaRNA package version 2.4.14[8,9], SimFold version 2.1[16,17], and RNAstructure version 6.2[10,11]. The parameters of four trainable algorithms, namely, MXfold, CONTRAfold, ContextFold and TORNADO, were trained with TrainSetA. For the other algorithms, default parameters were used.

Figures 3a and 3b show PPV-SEN plots of prediction accuracy for TestSetA and TestSetB, respectively. The maximum expected accuracy (MEA)-based secondary structure predictions, TOR-NADO, CONTRAfold and CentroidFold, can control PPV and SEN through the parameter $\gamma$, and therefore, their accuracies are shown as the curves for multiple $\gamma \in \{2^n | -5 \leq n \leq 10\}$.

For the predictions on TestSetA, our algorithm, MXfold2, achieved the best accuracy ($F = 0.761$), followed by ContextFold ($F = 0.759$). While the difference in $F$-value between MXfold2 and ContextFold was not significant ($p = 0.49$, one-sided Wilcoxon singed-rank test), MXfold2 was significantly more accurate than the other methods, including TORNADO ($F = 0.746$ at $\gamma = 4.0$), MXfold ($F = 0.739$), and CONTRAfold ($F = 0.719$ at $\gamma = 4.0$) ($p < 0.001$). On the other hand, for the

predictions on TestSetB, which is structurally dissimilar from TrainSetA, we observe that ContextFold achieved the worst accuracy ($F = 0.502$), consistent with the possible proneness of ContextFold to overfitting, as noted by Rivas et al.[15]. In contrast, we do not observe heavy overfitting, as occurred with Context-Fold, with MXfold2 ($F = 0.601$), which is also significantly more accurate than the other methods, including MXfold ($F = 0.581$), CONTRAfold ($F = 0.573$ at $\gamma = 4.0$), and TORNADO ($F = 0.552$ at $\gamma = 4.0$) ($p < 0.001$).

Figures 3c and 3d show the PPV-SEN plot and the distribution of $F$-values, respectively, on the combined dataset comprising TestSetA and TestSetB. These results indicate that MXfold2 ($F = 0.693$) achieved the best accuracy, followed by the trainable methods, namely, MXfold ($F = 0.673$), TORNADO ($F = 0.664$ at $\gamma = 4.0$), CONTRAfold ($F = 0.658$ at $\gamma = 4.0$), and ContextFold ($F = 0.651$), and MXfold2 also outperformed the thermodynamics-based methods ($p < 0.001$, one-sided Wilcoxon singed-rank test).

**Comparison with DNN-based methods**. We compared our algorithm, MXfold2, with two recently published DNN-based

methods: SPOT-RNA[19] and E2Efold[20]. We trained MXfold2 using the available SPOT-RNA and E2Efold training datasets and tested the prediction accuracy on their test datasets for direct comparison because SPOT-RNA does not provide training modules. Furthermore, because their datasets were designed only for sequence-wise cross-validation, for the purpose of family-wise cross-validation, we developed the bpRNA-new dataset, which does not contain any families present in their training datasets.

Table 2 and Supplementary Fig. 2 show the comparison of the prediction accuracy among MXfold2, SPOT-RNA, TORNADO, ContextFold, and RNAfold, with MXfold2 and ContextFold trained on the TR0 dataset, which is a subset of the bpRNA-1m dataset[24]. In contrast, SPOT-RNA used the pre-trained model trained on the TR0 dataset and an additional dataset for transfer learning. All of these methods were tested using the TS0 dataset, which is also a subset of the bpRNA-1m dataset, for sequence-wise cross-validation, after which the bpRNA-new dataset was used for family-wise cross-validation.

We observe that MXfold2 performs slightly worse in sequence-wise cross-validation relative to SPOT-RNA, but the difference is not significant ($p = 0.31$, one-sided Wilcoxon signed-rank test). On the other hand, in terms of family-wise cross-validation, MXfold2 is significantly more accurate than SPOT-RNA ($p < 0.001$). In addition, as shown in Supplementary Fig. 3, MXfold2

is, on average, approximately 15 and 36 times faster than SPOT-RNA when using GPU and CPU, respectively.

Figures 4 and 5 show typical examples of successful and failed predictions by MXfold2. We find in Fig. 4 that our method, which integrates the thermodynamic model, enables accurate predictions even when machine learning-based methods cannot be sufficiently trained. However, Fig. 5 suggests that if neither machine learning-based methods nor the thermodynamic-based method can accurately predict the secondary structure, MXfold2 may also be difficult to predict it successfully.

We also compared MXfold2 with E2Efold, TORNADO, ContextFold, and RNAfold (Supplementary Table 5 and Supplementary Fig. 4), in which E2Efold used the pre-trained model trained on a subset of the RNAStrAlign dataset[25], and MXfold2 and ContextFold were trained on the same dataset. All of these methods were tested on a subset of the ArchiveII dataset[26] for sequence-wise cross-validation and the bpRNA-new dataset for family-wise cross-validation. We observe that E2Efold almost completely failed to predict secondary structures for unseen families ($F = 0.0361$), whereas MXfold2 accurately predicted their secondary structures ($F = 0.628$). Furthermore, E2Efold trained with TrainSetA also failed to predict secondary structures for TestSetB ($F = 0.0322$), which is outside of Fig. 3. These results suggest that E2Efold might be categorized as being prone to heavy overfitting.

**Table 2 Comparison of the accuracy of secondary structure prediction among MXfold2, SPOT-RNA, TORNADO, ContextFold, and RNAfold.**

|  | Sequence-wise CV[a] | | | Family-wise CV[b] | | |
|---|---|---|---|---|---|---|
|  | PPV | SEN | F | PPV | SEN | F |
| MXfold2[c] | 0.520 | 0.682 | 0.575 | 0.585 | 0.710 | 0.632 |
| SPOT-RNA[c] | 0.652 | 0.578 | 0.597 | 0.599 | 0.619 | 0.596 |
| TORNADO[c] | 0.554 | 0.609 | 0.561 | 0.636 | 0.638 | 0.620 |
| ContextFold[c] | 0.583 | 0.595 | 0.575 | 0.595 | 0.539 | 0.554 |
| RNAfold | 0.446 | 0.631 | 0.508 | 0.552 | 0.720 | 0.617 |

[a]Sequence-wise cross-validation (CV) with the TS0 dataset.
[b]Family-wise CV with the bpRNA-new dataset.
[c]All trainable methods were trained with the TR0 dataset.

**Correlation with free energy**. We investigated the correlation between the free energy and the predicted folding score using the T-Full dataset[17], which contains sequence-structure-energy triplets. Table 3 shows the root mean square error (RMSE) and the Spearman's rank correlation coefficient ($\rho$) between the free energy of the reference structure and the folding score of the predicted structure for MXfold2 with and without thermodynamic regularization compared with CONTRAfold and RNAfold. The folding score of MXfold2 is highly correlated with the true free energy, though it is not as high as that of RNAfold. In contrast, even though MXfold2 without thermodynamic regularization is able to predict secondary structures as accurately as MXfold2 with thermodynamic regularization, its folding score is not highly correlated with the true free energy. This suggests that

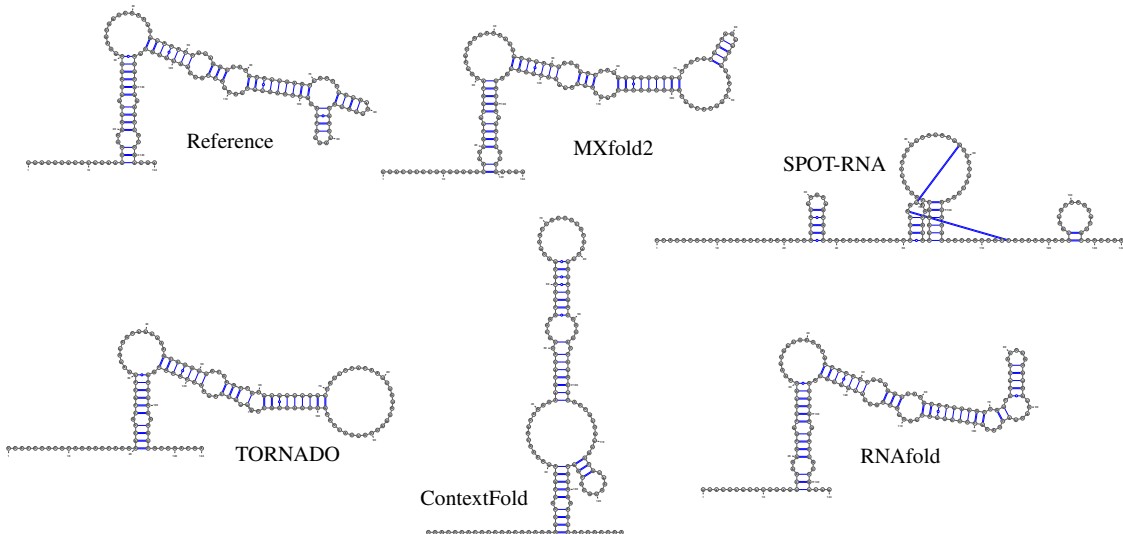

**Fig. 4 Comparison of predicted structures of** `FJ746361.1_1-144`**.** Here, we employed MXfold2 (PPV = 1.0, Sen = 0.85, F = 0.92), SPOT-RNA (PPV = 0.18, Sen = 0.37, F = 0.24), TORNADO (PPV = 0.967, Sen = 0.725, F = 0.829), ContextFold (PPV = 0.32, Sen = 0.23, F = 0.26), and RNAfold (PPV = 0.82, Sen = 0.80, F = 0.81).

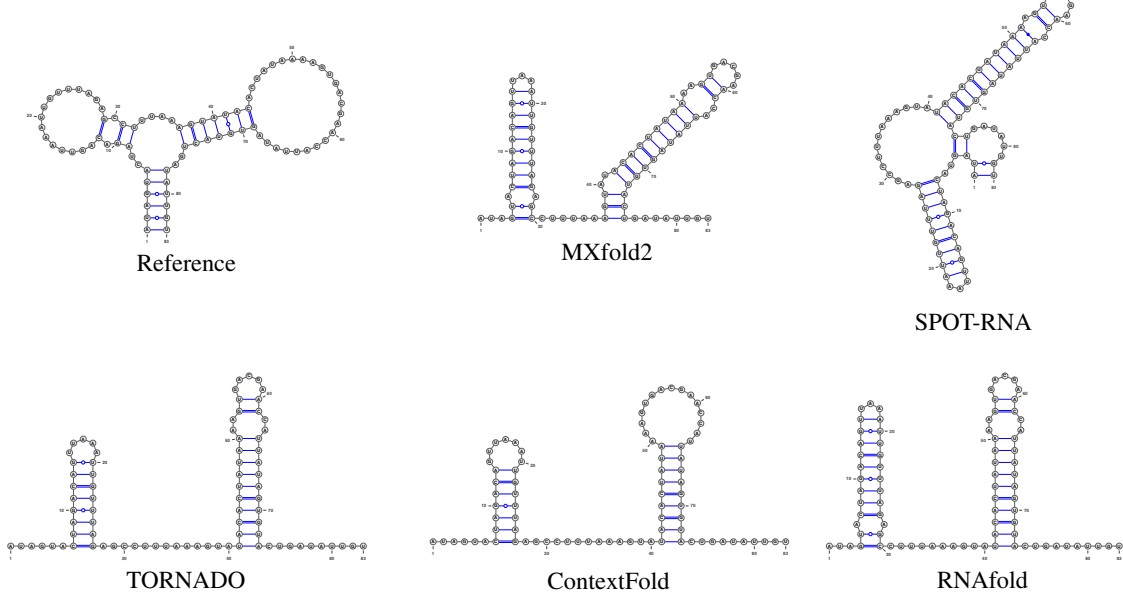

**Fig. 5 Comparison of predicted structures of** `URS0000D6A925_12908_1-83`**.** Here, we employed MXfold2 (PPV = 0.12, Sen = 0.19, F = 0.155), SPOT-RNA (PPV = 0.12, Sen = 0.08, F = 0.09), TORNADO (PPV = 0.0, Sen = 0.0, F = 0.0), ContextFold (PPV = 0.0, Sen = 0.0, F = 0.0), and RNAfold (PPV = 0.0, Sen = 0.0, F = 0.0).

**Table 3 Comparison of the correlation between predicted folding score and the reference free energy.**

|  | PPV | SEN | F | RMSE[a] | $\rho$[b] |
|---|---|---|---|---|---|
| MXfold2 | 0.984 | 0.978 | 0.980 | 3.260 | 0.833 |
| MXfold2 w/o th.[c] | 0.980 | 0.972 | 0.973 | 3.607 | 0.538 |
| CONTRAfold | 0.963 | 0.639 | 0.643 | 5.781 | 0.736 |
| RNAfold | 0.979 | 0.964 | 0.963 | 2.868 | 0.909 |

[a]The root mean square error measured in kcal/mol.
[b]The Spearman's rank correlation coefficient.
[c]MXfold2 without the thermodynamic regularization.

high accuracy in the secondary structure prediction does not directly lead to the reliable evaluation of thermodynamic stability.

## Discussion

We have proposed a deep learning-based algorithm for predicting RNA secondary structures with thermodynamic integration. Our experiments showed that thermodynamic integration, consisting of thermodynamic-integrated folding scores and thermodynamic regularization combined, substantially improved the robustness of predictions. We conducted benchmarks that compare our algorithm with conventional algorithms, using not only sequence-wise cross-validation, but also family-wise cross-validation, which assumes one of the most practical situations for RNA secondary structure prediction from single sequences. In this comparison, our algorithm achieved the best prediction accuracy without overfitting to the training data, as was the case with other rich-parameterized methods such as ContextFold and E2Efold. The proposed MXfold2 algorithm should be useful for improving RNA structure modeling, especially for newly discovered RNAs.

Our experiments showed that several methods with rich parameterization achieved excellent performance for test datasets structurally similar to training datasets, but poor performance for test datasets structurally dissimilar from training datasets. Rivas et al.[15] have already pointed out that rich-parameterized methods might easily tend towards overfitting. Such secondary structure prediction methods would be useful if we could know a priori

whether the sequence we want to predict is a member of a family included in the training data. However, if we have knowledge about the family of the sequences whose structures are to be predicted, homology searches for RNA sequences, such as Infernal[27], are a better choice. In other words, the lack of prior knowledge about sequences is a major reason the prediction of secondary structures from single sequences is an important goal. In principle, regardless of whether a given sequence is within a family included in a training dataset or not, robust methods should be able to properly predict its secondary structure because its secondary structure is formed under the same physical laws. In general, machine learning-based methods can be expected to improve prediction accuracy by increasing the number of training samples. Although the development of sequencing techniques has made it easier to obtain sequence data, it is still not easy to increase the amount of training data because of the significant effort required to obtain those empirically determined secondary structures. Therefore, it is impractical to predict secondary structures of newly discovered sequences using methods that are not generalizable to a wide variety of sequences, such as those methods that rely on only memorizing sequences and their secondary structures within a large number of neural network parameters. We designed our experiments to predict secondary structures of newly discovered RNAs using test datasets structurally dissimilar from training datasets, e.g., TestSetB for TrainSetA (originally developed by Rivas et al.[15]) and the bpRNA-new dataset for the TR0 dataset. These conditions are among the most important circumstances for RNA secondary structure prediction from single sequences, and our method, which is able to achieve high accuracy under these conditions, is useful in practice.

Deep neural network-based methods, including our method, improve prediction accuracy by optimizing a larger number of parameters for training data relative to conventional methods. For example, our algorithm employs 803k parameters, whereas SPOT-RNA and E2Efold use 1746k and 719k parameters, respectively. Therefore, as shown so far, such rich-parametrized models need to be trained more carefully because they are prone to overfitting. SPOT-RNA mitigates overfitting by building an ensemble of five deep models with transfer learning. To do that,

our MXfold2 integrates the folding scores, which are calculated by a deep neural network, with Turner's nearest-neighbor free energy parameters, and employs training using thermodynamic regularization. Whereas MXfold2 is limited to pseudoknot-free secondary structure prediction owing to its use of the nearest-neighbor model, SPOT-RNA and E2Efold are able to predict pseudoknotted structures because their RNA secondary structure prediction is formulated as multiple binary classification problems that predict whether each pair of nucleotides forms a base pair or not.

Estimation of free energies is important for applications other than structure prediction, such as small interfering RNA selection using hybridization thermodynamics[28]. Table 3 indicates that RNAfold and MXfold2 with thermodynamic regularization can calculate folding scores that are highly correlated with true free energy estimates, at least for sequences for which secondary structures can be predicted with high accuracy. Given that MXfold2 is more accurate in secondary structure prediction compared with RNAfold, as mentioned above, MXfold2 can be expected to evaluate thermodynamic stability with high accuracy for more sequences compared with RNAfold.

## Methods

**Using a deep neural network to compute folding scores.** Figure 2 shows the overview of our deep neural network, which is used to calculate four types of folding scores for each pair of nucleotides in a given sequence: helix stacking, helix opening, helix closing, and unpaired region.

The input of the network is an RNA sequence of length $L$. Each base (A, C, G, or U) is encoded into $d$-dimensional embeddings using a trainable embedding function.

The $L \times d$-dimensional sequence embedding is entered into 1D convolution blocks (Block 1) repeated $N_1$ times. Each 1D convolution block consists of two convolutional layers, with kernel sizes of 5 and 3, respectively, and a depth of $d$. Layer normalization[29] and continuously differentiable exponential linear units (CELU) activation functions[30] are employed. A dropout rate of 50% is used to avoid overfitting[31]. To efficiently train deep neural networks, a residual connection[32] that skips each layer is employed.

The next block (Block 2) contains $N_2$ layers of the bidirectional long short-term memory (BiLSTM) network[33]. The number of hidden units in each direction of the BiLSTM network is $d/2$. Each BiLSTM layer is followed by the layer normalization and CELU activation functions. A dropout rate of 50% is also used in BiLSTM layers.

The $L \times d$ matrix obtained so far is divided into two matrices of size $L \times d/2$; one of the two matrices expands in the column direction and the other in the row direction, and then the two matrices are concatenated into an $L \times L \times d$ matrix, as shown in the "Concat 2D" operation in Fig. 2. This matrix is entered into 2D convolution blocks (Block 3) $N_3$ times. Each 2D convolution block consists of two convolutional layers, each with a kernel size of $5 \times 5$ and $3 \times 3$, respectively, and a depth of $d$. The layer normalization and CELU functions with a dropout rate of 50% are again utilized.

Finally, four types of folding scores for each pair of $i$-th and $j$-th nucleotides are calculated by 3-layer multilayer perceptron (MLP) with $h$ hidden units.

We used the following values for the above hyperparameters: $d = 64$, $N_1 = 4$, $N_2 = 2$, $N_3 = 4$, and $h = 32$.

**Using DP to predicting RNA secondary structure.** Similar to traditional folding methods, such as RNAfold and CONTRAfold, our algorithm adopts the nearest-neighbor model[4,34] as an RNA secondary structure model without pseudoknots. As an alternative to the thermodynamic parameters, our algorithm calculates the free energy of the nearest-neighbor loops using four types of folding scores computed by the deep neural network described in the previous section. The four types of the folding scores are helix stacking scores, helix opening scores, helix closing scores, and unpaired region scores (Fig. 2). The helix stacking scores characterize the stacking of base-pairs that mainly contributes to the thermodynamic stability of RNA structures. The helix opening scores and helix closing scores characterize each end of helix regions, corresponding to the terminal mismatch in Turner's thermodynamic model. The unpaired region scores characterize unstructured sequences that contribute to the thermodynamic instability of RNA structures. Our algorithm calculates the folding score for a nearest-neighbor loop by summing up the four types of the scores comprising the loop. For example, the folding score of the internal loop consisting of bases 25–29 and 43–47 in Fig. 1 is calculated by summing the helix closing score for (25, 47), the helix opening score for (28, 44), and the unpaired region scores for (26, 28) and (44, 46). Similarly, the folding score of the base-pair stacking consisting of 23–24 and 48–49 in Fig. 1 is calculated from the helix stacking score for (23, 49).

Similar to MXfold, we integrate our model with the thermodynamic model by adding the negative value of the Turner free energy[4] to the folding scores calculated

by the deep neural network for each nearest-neighbor loop. We define our scoring function of a secondary structure $y$ given an RNA sequence $x$ as

$$f(x,y) = f_T(x,y) + f_W(x,y), \tag{1}$$

where $f_T(x,y)$ is the contribution of the thermodynamic parameters (i.e., the negative value of the free energy of $y$) and $f_W(x,y)$ is the contribution of the deep neural network, which is calculated as the sum of the folding scores over the decomposed nearest-neighbor loops as described above. The thermodynamic-integrated folding scores defined by Eq. (1) can also be decomposed into nearest-neighbor loops as each term of Eq. (1) can be decomposed in the same manner. Therefore, we can efficiently calculate an optimal secondary structure that maximizes the scoring function (Eq. 1) using the Zuker-style dynamic programming (DP) algorithm[5]. See Supplementary Methods for more details of the Zuker-style DP algorithm.

**Training parameters with the max-margin framework.** To optimize the network parameters $\lambda$, we employed a max-margin framework called the structured support vector machine (SSVM)[35]. Given a training dataset $\mathcal{D} = \{(x^{(k)}, y^{(k)})\}_{k=1}^{K}$, where $x^{(k)}$ is the $k$-th RNA sequence and $y^{(k)}$ is the reference secondary structure for the $k$-th sequence $x^{(k)}$, we aim to find a set of parameters $\lambda$ that minimizes the objective function

$$\mathcal{L}(\lambda) = \sum_{(x,y)\in\mathcal{D}} \left\{ \left( \max_{\hat{y}\in\mathcal{S}(x)} [f(x,\hat{y}) + \Delta(y,\hat{y})] - f(x,y) \right) + C_1 [f(x,y) - f_T(x,y)]^2 + C_2 \|\lambda\|_2 \right\}, \tag{2}$$

where $\mathcal{S}(x)$ is a set of all possible secondary structures of $x$. The first term is the structured hinge loss function[35]. Here, $\Delta(y,\hat{y})$ is a margin term of $\hat{y}$ for $y$ defined as

$$\Delta(y,\hat{y}) = \delta^{FN} \times (\text{# of false-negative base pairs})$$
$$+ \delta^{FP} \times (\text{# of false-positive base pairs}),$$

where $\delta^{FN}$ and $\delta^{FP}$ are tunable hyperparameters that control the trade-off between sensitivity and specificity for learning the parameters. We used $\delta^{FN} = 0.5$ and $\delta^{FP} = 0.005$ by default. The margin term for structured models enables robust predictions by maximizing the margin between $f(x,y)$ and $f(x,\hat{y})$ for $y \neq \hat{y}$. We can calculate the first term of Eq. (2) using Zuker-style dynamic programming[5] modified by the use of loss-augmented inference[35]. The second and third terms of Eq. (2) are regularization terms that penalize parameter values that are extreme relative to a postulated distribution. The second term is our proposed thermodynamic regularization, which prevents the folding score of the secondary structure from differing significantly from the free energy of the thermodynamic parameters. The third term of Eq. (2) is the $\ell_2$ regularization term. We used $C_1 = 0.125$ and $C_2 = 0.01$ by default. To minimize the objective function (2), we employ the Adam optimizer[36].

**Datasets.** To evaluate our algorithm, we performed computational experiments on several datasets. Supplementary Table 1 shows a summary of the datasets used in our experiments.

The first dataset, which has been carefully established by Rivas et al.[15], includes TrainSetA, TestSetA, TrainSetB, and TestSetB. TrainSetA and TestSetA were collected from the literature[12,16,17,37–39]. TrainSetB and TestSetB, which contain 22 families with 3D structure annotations, were extracted from Rfam 10.0[40]. The sequences in Train/TestSetB share less than 70% sequence identity with the sequences in TrainSetA. We excluded a number of sequences that contain pseudoknotted secondary structures in the original data sources from all four sub-datasets since all algorithms evaluated in this study were designed for predicting RNA secondary structures without pseudoknots. It is important to note that literature-based TrainSetA and Rfam-based TestSetB are structurally dissimilar, whereas TrainSetA and TestSetA have some structural similarity.

To compare our algorithm with SPOT-RNA[19], we employed the same dataset extracted from the bpRNA-1m dataset[24], which is based on Rfam 12.2[41] with 2588 families. After removing redundant sequences based on sequence identity, the bpRNA-1m dataset was randomly split into three sub-datasets, namely, TR0, VL0, and TS0, for training, validation, and testing, respectively. Note that this partition is not family-wise, but sequence-wise. SPOT-RNA performed the initial training and validation using TR0 and VL0, which was followed by testing using TS0 and then transfer learning using other PDB-based datasets.

To confirm the robustness against "unseen" families, we built a dataset that includes families from the most recent Rfam database, Rfam 14.2[42]. Since the release of Rfam 12.2, from which bpRNA-1m is derived, the Rfam project has been actively collecting about 1,500 RNA families, including families detected by newly developed techniques[43]. We first extracted these newly discovered families. Then, as with SPOT-RNA, we removed redundant sequences by CD-HIT-EST[44], with a cutoff threshold of 80%, and discarded sequences whose lengths exceeded 500 bp. Our dataset is referred to as bpRNA-new.

We also compared our algorithm with E2Efold[20], using the same experimental conditions in which the RNAStrAlign dataset[25] and the ArchiveII dataset[26] were employed for training and testing, respectively. Note that this setting is also not family-wise, but sequence-wise, because both datasets contain the same families.

To investigate the correlation between folding score and free energy, we employed the T-Full dataset, which was compiled by Andronescu et al.[17].

Each sequence in the T-Full dataset has a corresponding reference secondary structure, as well as its free energy change derived from optical melting experiments.

**Performance measure**. We evaluated the accuracy of predicting RNA secondary structures through the positive predictive value (PPV) and the sensitivity (SEN) of the base pairs, defined as

$$\text{PPV} = \frac{\text{TP}}{\text{TP} + \text{FP}}, \quad \text{SEN} = \frac{\text{TP}}{\text{TP} + \text{TN}},$$

where TP is the number of correctly predicted base pairs (true positives), FP is the number of incorrectly predicted base pairs (false positives), and FN is the number of base pairs in the reference structure that were not predicted (false negatives). We also used the $F$-value as the balanced measure between PPV and SEN, which is defined as their harmonic mean:

$$F = \frac{2 \times \text{SEN} \times \text{PPV}}{\text{SEN} + \text{PPV}}.$$

For the experiment on the T-Full dataset, we evaluated the root mean square error (RMSE)

$$\text{RMSE} = \sqrt{\frac{1}{N} \sum_{i=1}^{N} (e_i - \hat{e}_i)^2},$$

where $e_i$ and $\hat{e}_i$ are the free energy of the reference secondary structure and the negative value of the folding score of the predicted secondary structure for the $i$-th sequence, respectively. We also evaluated the Spearman's rank correlation coefficient ($\rho$) between $e$ and $\hat{e}$.

**Reporting summary**. Further information on research design is available in the Nature Research Reporting Summary linked to this article.

## Data availability

The datasets used in this study are available at https://doi.org/10.5281/zenodo.4430150.

## Code availability

The MXfold2 source code is available at https://github.com/keio-bioinformatics/mxfold2/, and the MXfold2 web server is available for use at http://www.dna.bio.keio.ac.jp/mxfold2/.

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

## Acknowledgements

This work was partially supported by a Grant-in-Aid for Scientific Research (B) (No. 19H04210) and Challenging Research (Exploratory) (No. 19K22897) from the Japan Society for the Promotion of Science (JSPS) to K.S. and a Grant-in-Aid for JSPS Fellows (No. 18J21767) from JSPS to M.A. This work was also supported by a Grant-in-Aid for Scientific Research on Innovative Areas "Frontier Research on Chemical Communications." The supercomputer system used for this research was provided by the National Institute of Genetics (NIG), Research Organization of Information and Systems (ROIS).

## Author contributions

K.S. conceived the study, implemented the algorithm, conducted experiments, and drafted the manuscript. K.S. and M.A. discussed the network architectures, designed the experiments, and collected the datasets. K.S. and Y.S. discussed the training algorithm. All authors read, contributed to the discussion of, and approved the final manuscript.

## Competing interests

The authors declare no competing interests.
