## [Peer Review File · Nature Communications]

Reviewers' Comments:

Reviewer #1:

Remarks to the Author:

The manuscript describes a method for RNA secondary structure prediction. The method integrates the recent deep learning approaches (SPOT-RNA and E2Efold) with the thermodynamic energy parameters used by the authors in their previous work (MXfold). To integrate the two, the authors propose thermodynamic regularization in the network training. While the addition of thermodynamic parameters and regularization improves the accuracy compared to the base model (0.46 vs. 0.57) on the RNA structures that are not part of the training set (TestSetB), there is still a huge gap between the TestSetA (RNAs with similarity to training set) and TestSetB (0.75 vs. 0.57). This gap indicates overfitting still occurs. Overall the method makes progress in the right direction, but more work is needed to prevent overfitting.

Minor:

Please provide the total number of network parameters and how it compares to SPOR-RNA or E2Efold networks).

Analyze/include examples of RNAs where the prediction fails.

In the abstract please provide numbers that show how accurate MXfold2 is compared to other algorithms.

There are two almost identical sentences at the end of the abstract

Introduction: cryoEM is emerging as a technique for experimental structure determination of RNA (such as Kappel et al. 2020). Please cite the recent work on it.

Reviewer #2:

Remarks to the Author:

This manuscript presents a new RNA folding algorithm for a single RNA sequence. The algorithm combines thermodynamic scores with scores from a neural network that classifies pairs into four categories: an opening pair, a closing pair, an internal base pair, or unpaired. This integration of two types of scores has been introduced before by the same authors in a method called MXfold. In addition, this new method (named MXfold2) introduces a regularization based on the thermodynamic scores for the neural network classification that is meant to avoid overfitting. The authors show results comparing to other methods on known Training/Test sets designed to test overfitting to specific RNA structures.

The method is interesting. The method is poorly described (see specific comments below). The results are acceptable, but they need to be put better into context, both with other methods (the NN network models prior to this, and also probabilistic models), and with similar previous results with the same benchmarks.

Specific comments.

- The names used for the different techniques in this work are confusing, overlapping and not very well explained. I believe the authors use three concepts:

(1) "thermodynamic-integrated folding scores". This method seems to be described in their previous method MXfold. I find it impossible to get the pdf of this

published work, even through my University. Thus, I believe a better explanation is required.

In addition this concept is also called "thermodynamic integrations", for instance here in the abstract

"the folding scores are integrated with traditional thermodynamic parameters to enable robust predictions."

but the term "thermodynamic integration" is later defined again as the final product of work (see (3)).

(2) "Thermodynamic regularization" (the contribution of this work, as the authors themselves state).

Please explain in more detail. All I can find is

"The deep neural network is trained by the max-margin framework with thermodynamic regularization" (page 4)

If this is the major result of this work, it deserved a better description in the main text.

(3) "Thermodynamic integration" (the combination of the two above)

"These two techniques, the thermodynamic-integrated folding scores and the thermodynamic regularization, are referred to as thermodynamic integration."

But the name "integration" has been used before for (1) (for instance in abstract)

"the folding scores are integrated with traditional thermodynamic parameters to enable robust predictions."

- When the authors describe alternative approaches in page 3, they seem to forget to mention probabilistic methods, such as Conus or Tornado. I am also missing at least one probabilistic method to be included in the benchmark. Rivas et al 20 includes several probabilistic methods trained/tested using the same benchmarks that the authors use here. (See more below about this when I discuss Figure 3).

- The DNN architecture needs a better description. Many terms in Table 1 are not defined, for instance:
d, N1, N2, BiLSTM, CONCAT 2D, MPL

Also, describe with some detail what the 4 scores derived from the DNN are.

- A "Zuker-style" dynamic programming algorithm can probably be describe with some more precision.

- The authors compare their method to other methods in Figure 3. I am missing in that same figure a comparison to the other two methods that used a NN Comparison to SPOT-RNA and E2EFOLD.

- In comparing the results presented in Figure 3 with those reported in Figure 4 of Rivas et al., 2012, some discrepancies are evident. The training/testing sets are the same, yet most methods seem to perform better in Figure 3 than in Rivas et al. 2012.

These differences could happen because the methods have improved over time. That does not seem to be the case for ContextFold, for instance.

There are important differences in the numbers reported in Rivas et al 2012 and those of Figure 3 for ContexFold. Rivas et al. 2012 reports using ContexFold version v1.0. I just downloaded ContexFold from <https://www.cs.bgu.ac.il/~negevcb/contextfold/> and the version is still 1.0.

Another reason could be the quantity reported are different. In Rivas 2020, the total sen/ppv are reported. It is unclear whether Figure 3 reports total values or average values, or something else. Considering the large variability of performance from sequence to sequence (reported in Figure 3d), using total versus mean values could result in large differences.

Independently of the differences with previous publications, it is important to notice that the differences for the different methods are not very meaningful in the context of the large spread of results for different RNA sequences.

For instance, if we were to add a confidence range to Figures 3a, 3b or 3c, none of the differences would look significant. That is an issue larger than this particular method, and I would not expect any particular method to give much lower variance from the same input (just the single RNA sequence). Nevertheless, it is important to express clearly the limitations of this approach.

The versions of all programs used should be provided in order to be able to reproduce results.

- In Figure 3, please the same scales for all PPV/sen plots

Answers to Reviewer 1's comments

Thank you very much for reading our manuscript carefully and providing helpful comments.

The manuscript describes a method for RNA secondary structure prediction. The method integrates the recent deep learning approaches (SPOT-RNA and E2Efold) with the thermodynamic energy parameters used by the authors in their previous work (MXfold). To integrate the two, the authors propose thermodynamic regularization in the network training. While the addition of thermodynamic parameters and regularization improves the accuracy compared to the base model (0.46 vs. 0.57) on the RNA structures that are not part of the training set (TestSetB), there is still a huge gap between the TestSetA (RNAs with similarity to training set) and TestSetB (0.75 vs. 0.57). This gap indicates overfitting still occurs. Overall the method makes progress in the right direction, but more work is needed to prevent overfitting.

Answer. We greatly appreciate your kind comments. Based on these comments, we have revised our paper as indicated below.

We agree that overfitting has not been completely overcome. As you pointed out, the gap in prediction accuracy for our method between TestSetA and TestSetB is large (0.76 vs. 0.60 in F-value). However, this gap is also quite large for RNAfold (0.64 vs. 0.54), which does not need training, suggesting that TestSetB is a more difficult sequence group for the prediction of secondary structures relative to TestSetA and that the large difference in accuracy is not explained by overfitting alone.

Please provide the total number of network parameters and how it compares to SPOT-RNA or E2Efold networks.

Answer. Thank you for the helpful suggestion. We have added the following paragraph including the number of parameters and the comparison to SPOT-RNA and E2Efold in the Discussion section on p. 11:

Deep neural network-based methods, including our method, improve prediction accuracy by optimizing a larger number of parameters for training data relative to conventional methods. For example, our algorithm employs 803k parameters, whereas SPOT-RNA and E2Efold use 1,746k and 719k parameters, respectively. Therefore, as shown so far, such rich-parametrized models need to be trained more carefully because they are prone to overfitting. SPOT-RNA mitigates overfitting by building an ensemble of five deep models with transfer learning. To do that, our MXfold2 integrates the folding scores, which are calculated by a deep neural network, with Turner's nearest-neighbor free energy parameters, and employs training using thermodynamic regularization. Whereas MXfold2 is limited to pseudoknot-free secondary structure prediction owing to its use of the nearest neighbor model, SPOT-RNA and E2Efold are able to predict pseudoknotted structures because their RNA secondary structure prediction is formulated as multiple binary classification problems that predict whether each pair of nucleotides forms a base pair or not.

Analyze/include examples of RNAs where the prediction fails.

Answer. We have added a typical example of predictions of our method as Figure 4, and the paragraph in the Result section on p. 9 as follows:

Figure 4 shows a typical example of successful predictions by MXfold2. We find that our method, which integrates the thermodynamic model, enables accurate predictions even when machine learning-based methods cannot be sufficiently trained.

In the abstract please provide numbers that show how accurate MXfold2 is compared to other algorithms.

Answer. Thank you for the suggestion. We have rewritten the last sentence of the abstract as follows:

Our algorithm (MXfold2) achieved the most robust and accurate predictions in computational experiments designed for newly discovered non-coding RNAs, with significant 2–10 % improvements over our previous algorithm (MXfold) and standard algorithms for predicting RNA secondary structures in terms of F -value.

There are two almost identical sentences at the end of the abstract

Answer. We removed one of the duplicated sentences.

Introduction: cryoEM is emerging as a technique for experimental structure determination of RNA (such as Kappel et al. 2020). Please cite the recent work on it.

Answer. Thank you for the valuable suggestion. We have added the following sentence to the Introduction, on p. 2:

Although considerable advances in cryo-electron microscopy research on RNA tertiary structure determination have been achieved in recent years, these limitations have not yet been completely overcome.

with the citation to Kappel et al. (2020).

Answers to Reviewer 2's comments

Thank you very much for reading our manuscript carefully and providing helpful comments.

This manuscript presents a new RNA folding algorithm for a single RNA sequence. The algorithm combines thermodynamic scores with scores from a neural network that classifies pairs into four categories: an opening pair, a closing pair, an internal base pair, or unpaired. This integration of two types of scores has been introduced before by the same authors in a method called MXfold. In addition, this new method (named MXfold2) introduces a regularization based on the thermodynamic scores for the neural network classification that is meant to avoid overfitting. The authors show results comparing to other methods on known Training/Test sets designed to test overfitting to specific RNA structures.

The method is interesting. The method is poorly described (see specific comments below). The results are acceptable, but they need to be put better into context, both with other methods (the NN network models prior to this, and also probabilistic models), and with similar previous results with the same benchmarks.

Answer. We greatly appreciate your kind comments. Based on these comments, we have revised our paper as described below.

The names used for the different techniques in this work are confusing, overlapping and not very well explained. I believe the authors use three concepts:

Answer. We apologize for the confusion about the terminology used to describe our proposed techniques. We have attempted to clarify this issue as described below.

(1) "thermodynamic-integrated folding scores". This method seems to be described in their previous method MXfold. I find it impossible to get the pdf of this published work, even through my University. Thus, I believe a better explanation is required.

In addition this concept is also called "thermodynamic integrations", for instance here in the abstract

"the folding scores are integrated with traditional thermodynamic parameters to enable robust predictions."

but the term "thermodynamic integration" is later defined again as the final product of work (see (3)).

Answer. We have added an explanation of "thermodynamic-integrated folding scores" on p. 15 as follows:

The thermodynamic-integrated folding scores defined by Eq. (1) can also be decomposed into nearest-neighbor loops as each term of Eq. (1) can be decomposed in the same manner. Therefore, we can efficiently calculate an optimal secondary structure that

maximizes the scoring function (1) using the Zuker-style dynamic programming (DP) algorithm.

In the previous manuscript, we used the term “thermodynamic integration” to describe the combination of “thermodynamic-integrated folding scores” and “thermodynamic regularization.” This terminology may certainly be confusing. Therefore, we have decided not to use the term “thermodynamic integration” as a collective term for the two techniques as much as possible.

(2) “Thermodynamic regularization” (the contribution of this work, as the authors themselves state).

Please explain in more detail. All I can find is “The deep neural network is trained by the max-margin framework with thermodynamic regularization” (page 4)

If this is the major result of this work, it deserved a better description in the main text.

Answer. Thank you for the suggestion. Thermodynamic regularization is explained on p. 21 in the Method section. To provide more detail in the Introduction, we have modified the above sentence as follows:

The deep neural network is trained using the max-margin framework with thermodynamic regularization, which is proposed in this paper to make our model robust by maintaining that the folding score calculated by our method and the free energy calculated by the thermodynamic parameters are as close as possible.

Furthermore, we modified the last sentence of the “Overview of our algorithm” subsection as follows:

The deep neural network is trained using the max-margin framework, also known as structured support vector machine (SSVM), to minimize the structured hinge loss function with thermodynamic regularization, which prevents the folding score of the secondary structure from differing significantly from the free energy of the thermodynamic parameters.

(3) “Thermodynamic integration” (the combination of the two above)

“These two techniques, the thermodynamic-integrated folding scores and the thermodynamic regularization, are referred to as thermodynamic integration.”

But the name “integration” has been used before for (1) (for instance in abstract)

“the folding scores are integrated with traditional thermodynamic parameters to enable robust predictions.”

Answer. As mentioned above, we have rewritten our manuscript to avoid using the term “thermodynamic integration”, except in the title. Instead, the term “(two) thermodynamic-related techniques” has been used for this purpose on p. 5 and p. 6 and the caption of Table 1.

When the authors describe alternative approaches in page 3, they seem to forget to mention probabilistic methods, such as Conus or Tornado. I am also missing at least one probabilistic method to be included in the benchmark. Rivas et al 20 includes several probabilistic methods trained/tested using the same benchmarks that the authors use here. (See more below about this when I discuss Figure 3).

Answer. We have added Tornado to Fig. 3. In addition, we have added the following sentences about probabilistic models in the Introduction section:

Probabilistic generative models such as stochastic context-free grammars (SCFGs) have also been applied to predicting RNA secondary structures. Recently, Tornado implemented an application of SCFGs to the nearest neighbor model, thus achieving performance comparable with that of other machine learning-based methods.

The DNN architecture needs a better description. Many terms in Table 1 are not defined, for instance: d , N_1 , N_2 , BiLSTM, CONCAT 2D, MPL

Also, describe with some detail what the 4 scores derived from the DNN are.

Answer. We have modified the caption of Figure 2 as follows to describe terms:

The network structure of our algorithm that calculates four types of folding scores for each pair of nucleotides (i, j) from a given RNA sequence of length L . See the METHODS section for more details. act.: the activation function, norm.: the normalization function, conv: convolution layer, d : the dimension of trainable embeddings, N_1 , N_2 , and N_3 : the numbers of times Blocks 1, 2, and 3 repeat, respectively, BiLSTM: bidirectional long short-term memory, MLP: multilayer perceptron, and Concat 2D: concatenation of 1D sequences to 2D matrices.

In addition, we added definitions for the terms BiLSTM, MLP, and, “Concat 2D” to the Methods section.

To describe the four types of folding scores in greater detail, we added the following paragraph in the Method section:

The helix stacking scores characterize the stacking of base-pairs that mainly contributes to the thermodynamic stability of RNA structures. The helix opening scores and helix closing scores characterize each end of helix regions, corresponding to the terminal mismatch in Turner’s thermodynamic model. The unpaired region scores characterize unstructured sequences that contribute to the thermodynamic instability of RNA structures. Our algorithm calculates the folding score for a nearest-neighbor loop by summing up the four types of the scores comprising the loop.

A “Zuker-style” dynamic programming algorithm can probably be describe with some more precision.

Answer. To show more clearly the relationship between the Zuker algorithm and the thermodynamic models, we have added the following sentence to the Introduction: “We can efficiently calculate an optimal secondary structure that has the minimum free energy using a dynamic programming (DP) technique, the well-known Zuker algorithm.” Furthermore, we have also added citations of Zuker et al. (1981) with the use of the phrase “Zuker-style dynamic programming” in three places.

The authors compare their method to other methods in Figure 3. I am missing in that same figure a comparison to the other two methods that used a NN Comparison to SPOT-RNA and E2EFOLD.

Answer. Thank you for the valuable suggestion. We considered including DNN-based methods in Fig. 3, but unfortunately abandoned this objective because currently only SPOT-RNA provides the trained parameters trained by their dataset, but no training modules. We added the phrase “because SPOT-RNA does not provide training modules” to the section “Comparison with DNN-based methods”.

E2Efold provides training modules for user datasets. However, we decided that E2Efold does not need to be added to Fig. 3, because E2Efold trained with TrainSetA has an F-value of 0.03 for TestSetB, which contrasts strongly with the values shown in Fig. 3.

In comparing the results presented in Figure 3 with those reported in Figure 4 of Rivas et al., 2012, some discrepancies are evident. The training/testing sets are the same, yet most methods seem to perform better in Figure 3 than in Rivas et al. 2012.

Answer. Thank you for raising this important point. We expect this difference to arise for two reasons, as follows.

These differences could happen because the methods have improved over time. That does not seem to be the case for ContextFold, for instance. There are important differences in the numbers reported in Rivas et al 2012 and those of Figure 3 for ContextFold. Rivas et al. 2012 reports using ContextFold version v1.0. I just downloaded ContextFold from <https://www.cs.bgu.ac.il/negevcb/contextfold/> and the version is still 1.0.

Answer. We do not believe there are any significant improvements over time for either program. Instead, as we stated in the “Comparison with the existing methods” subsection, we trained CONTRAfold and ContextFold with TrainSetA in our experiments, although Rivas et al. (2012) used the default parameters trained with other datasets that are a subset of TrainSetA. This is the first factor explaining the difference in accuracy between the methods of Rivas et al. (2012) and our manuscript.

Another reason could be the quantity reported are different. In Rivas 2012, the total sen/ppv are reported. It is unclear whether Figure 3 reports total values or average values, or something else. Considering the large variability of performance from sequence to sequence (reported in Figure 3d), using total versus mean values could result in large differences.

Answer. The second factor explaining the difference in the accuracy is the calculation of sen/ppv, as you pointed out. We used the mean sen/ppv because many other papers other than Rivas et al. (2012) have used the mean sen/ppv. Rivas et al. (2012) pointed out problems in the use of mean ppv/sen, such as length bias, but we do not believe that the total ppv/sen completely resolves them. Instead, we provided the distribution of F-values for each method, as shown in Fig. 3d.

Independently of the differences with previous publications, it is important to notice that the differences for the different methods are not very meaningful in the context of the large spread of results for different RNA sequences.

For instance, if we were to add a confidence range to Figures 3a, 3b or 3c, none of the differences would look significant. That is an issue larger than this particular method, and I would not expect any particular method to give much lower variance from the same input (just the single RNA sequence). Nevertheless, it is important to express clearly the limitations of this approach.

Answer. Thank you for the important comment. As you note, the confidence ranges in the accuracy distribution are so wide that the significance is difficult to deduce from the figures. Therefore, we performed Wilcoxon signed-rank tests for the predictions on TestSetA, TestSetB, and the combined dataset to show the significance of the differences in accuracy between the methods. We have added a description of the test on p. 7 and p. 8, and the detailed results of the Wilcoxon signed-rank tests are shown in Supplementary Table 2 in the supplementary materials.

The versions of all programs used should be provided in order to be able to reproduce results.

Answer. We have already provided the versions of all programs used in our experiments in the “Comparison with the existing methods” subsection.

In Figure 3, please the same scales for all PPV/sen plots

Answer. Following your comment, we have made all the plots consistent in scale.

Reviewers' Comments:

Reviewer #1:

Remarks to the Author:

The authors addressed all the reviewer comments.

Reviewer #2:

Remarks to the Author:

The authors have improved the presentation of their algorithm and results. Bellow are a few comments regarding the updates introduced in the manuscript.

In response to one of my comments about all methods not performing significantly better than the others, and the high variability in performance from sequence to sequence, the authors have introduced a Wilcoxon signed-rank test. I think that is not the proper test to use. Wilcoxon provides a significant testes for the *means*. Comparing means is irrelevant. Sure you can see in Figure 3d that the means are different. The relevant question is how the performance change from sequence to sequence. Is one method consistently better for all sequences?

A better test is to emulate the graphs presented for AlphaFold2 in the casp-14 competition (attached). All sequences are ranked and the F values of each one of them is compared between two methods. I would do that for the author's to methods (for which I would expect maybe a correlation), and between their best method with the best other method, Tornado. I think this kind of depiction of the data would be really informative.

- Figure 3: Tornado as contrafold produce a sen/ppv curve. Not sure how the reported F for Tornado is produced. Please, report the curve.

- Since Tornado seems to be the best performing method after the MXfold and MXfold2 introduced by the authors, I think it is relevant that all figures and tables are updated to include Tornado results. That includes:

* Figures 4 and 5 to in main text

* Supplemental figures 1,2,3,4 and supplemental table 3.

- Figure 3d. Fix the position of the labels for the x-axis

Answers to Reviewer 2's comments

Thank you very much for reading our manuscript carefully and providing helpful comments.

The authors have improved the presentation of their algorithm and results. Below are a few comments regarding the updates introduced in the manuscript.

In response to one of my comments about all methods not performing significantly better than the others, and the high variability in performance from sequence to sequence, the authors have introduced a Wilcoxon signed-rank test. I think that is not the proper test to use. Wilcoxon provides a significant test for the *means*. Comparing means is irrelevant. Sure you can see in Figure 3d that the means are different. The relevant question is how the performance change from sequence to sequence. Is one method consistently better for all sequences?

A better test is to emulate the graphs presented for AlphaFold2 in the casp-14 competition (attached). All sequences are ranked and the F values of each one of them is compared between two methods. I would do that for the author's two methods (for which I would expect maybe a correlation), and between their best method with the best other method, Tornado. I think this kind of depiction of the data would be really informative.

Answer. Thank you for the valuable comment. We have added Supplementary Figure 5 that compares between MXfold2 and each competitive method for each sequence in the Φ values, which is similar with the AlphaFold2 graphs. Since it is difficult to show the superiority of MXfold2 just by looking at these plots, a statistical test is still necessary. We believe that the Wilcoxon signed-rank test is a non-parametric statistical hypothesis test used for comparing two related samples, or matched samples, not for comparing means. The Wilcoxon signed-rank test has been used for benchmarks for various applications including multiple sequence alignments (e.g. <https://doi.org/10.1038/srep33964>, <https://doi.org/10.1093/nar/gkz342>). Therefore, we think that the Wilcoxon signed-rank test is an appropriate test for this purpose.

Figure 3: Tornado as contrafold produce a sen/ppv curve. Not sure how the reported F for Tornado is produced. Please, report the curve.

Answer. We have added a PPV/SEN curve for TORNADO in Figure 3. The accuracies of TORNADO written in the main text is the accuracies at $\gamma = 4.0$. We have added this description on p. 7.

Since Tornado seems to be the best performing method after the MXfold and MXfold2 introduced by the authors, I think it is relevant that all figures and tables are updated to include Tornado results. That includes:

- Figures 4 and 5 to in main text
- Supplemental figures 1,2,3,4 and supplemental table 3.

Answer. Thank you for the suggestion. We have added the results of TORNADO to Figures 4 and 5, Supplementary Figures 2, 3, 4, and Supplementary Table 3. Because the aim of Supplementary Figure 1 is the comparison among variants of MXfold2, we have not added TORNADO to it.

Figure 3d. Fix the position of the labels for the x-axis

Answer. Fixed.

Reviewers' Comments:

Reviewer #2:

Remarks to the Author:

I am satisfied with the inclusion of supplemental Figure 5, which I believe shows the most accurate comparison of all presented models for single-sequence RNA structure prediction. That is, that overall there are large variations from method to method for a given sequence, and from sequence to sequence for a given method. And that there is no systematic way to decide one method or another performing better for any particular class of sequences.

Regarding the Wilcoxon test, given two distributions, it only looks at whether their population means can be call significantly different or not. Again that is not the relevant test for comparing methods that predict an RNA structure from a single sequence where the distribution performance is so spread, and where the mean has little relation with the performance of the method on any given RNA sequence. The fact that others have used the Wilcoxon test for similar problems is not argument to show that it is an appropriate method.

Answers to Reviewer 2's comments

Thank you very much for reading our manuscript carefully and providing helpful comments.

I am satisfied with the inclusion of supplemental Figure 5, which I believe shows the most accurate comparison of all presented models for single-sequence RNA structure prediction. That is, that overall there are large variations from method to method for a given sequence, and from sequence to sequence for a given method. And that there is no systematic way to decide one method or another performing better for any particular class of sequences.

Answer. We agree with your comment that there are large variations for each method, as shown in Fig 3d and Supplementary Fig 5.

Regarding the Wilcoxon test, given two distributions, it only looks at whether their population means can be call significantly different or not. Again that is not the relevant test for comparing methods that predict an RNA structure from a single sequence where the distribution performance is so spread, and where the mean has little relation with the performance of the method on any given RNA sequence. The fact that others have used the Wilcoxon test for similar problems is not argument to show that it is an appropriate method.

Answer. There are two 'Wilcoxon tests': the Wilcoxon rank-sum test and the Wilcoxon signed-rank test. The Wilcoxon rank-sum test, also known as the Mann-Whitney U test, is a non-parametric test for comparing two unpaired groups of samples. The Wilcoxon rank-sum test is inappropriate for this problem because this is not a paired test.

The Wilcoxon signed-rank test, which is employed in our paper, is a non-parametric test for comparing two groups of *paired* samples. Here, let a_i and b_i be the prediction accuracy (e.g. F values) by two methods, A and B, respectively, for i -th sequence in a given dataset that contains n sequences. We build two groups of paired samples, $a = \{a_1, \dots, a_n\}$ and $b = \{b_1, \dots, b_n\}$, where a_i and b_i are paired since both a_i and b_i are the accuracy for the i th sequence. Next, we calculate $|b_i - a_i|$ and their ranks from smallest to largest, say, if $|b_i - a_i|$ is smallest, it's rank $R_i = 1$. Then, we calculate the test statistics

$$W = \sum_{i=1}^n \text{sgn}(b_i - a_i) \cdot R_i,$$

where

$$\text{sgn}(x) = \begin{cases} 1 & (x > 0) \\ 0 & (x = 0) \\ -1 & (x < 0). \end{cases}$$

If there is no significant difference between a and b (null hypothesis), the mean and variance of W are 0 and $\frac{n(n+1)(2n+1)}{6}$, respectively. For $n \in \mathbb{N}$, W is normally distributed. We can see that the

Wilcoxon signed-rank test does not compare $\text{mean}(a)$ and $\text{mean}(b)$. Intuitively, the Wilcoxon signed-rank test is testing whether there is a significant difference between the number of A's won and the number of B's won. When we choose between the method A and method B for predicting secondary

structure of a given sequence, it is reasonable to choose the method with significantly more wins in the benchmark dataset.

In our experiments, no significant difference would be detected by comparing only the means or distributions using the unpaired test like the Wilcoxon rank-sum test since there are large variations for each method. Instead, since there is a correspondence between the groups by each sequence, it is appropriate to test by paired tests like the Wilcoxon signed-rank test.

Reviewers' Comments:

Reviewer #2:

None